# Spontaneous formation of nanoparticles on electrospun nanofibres

Norbert Radacsi [1,2], Fernando Diaz Campos[3], Calum R.I. Chisholm[3] & Konstantinos P. Giapis[1]

We report the spontaneous formation of nanoparticles on smooth nanofibres in a single-step electrospinning process, as an inexpensive and scalable method for producing high-surface-area composites. Layers of nanofibres, containing the proton conducting electrolyte, caesium dihydrogen phosphate, are deposited uniformly over large area substrates from clear solutions of the electrolyte mixed with polymers. Under certain conditions, the normally smooth nanofibres develop caesium dihydrogen phosphate nanoparticles in large numbers on their external surface. The nanoparticles appear to originate from the electrolyte within the fibres, which is transported to the outer surface after the fibres are deposited, as evidenced by cross-sectional imaging of the electrospun fibres. The presence of nanoparticles on the fibre surface yields composites with increased surface area of exposed electrolyte, which ultimately enhances electrocatalytic performance. Indeed, solid acid fuel cells fabricated with electrodes from processed nanofibre-nanoparticle composites, produced higher cell voltage as compared to fuel cells fabricated with state-of-the-art electrodes.

[1] Division of Chemistry and Chemical Engineering, California Institute of Technology, 1200 E. California Blvd, Pasadena, CA 91125, USA. [2] Institute for Materials and Processes, The School of Engineering, The University of Edinburgh, Robert Stevenson Road, Edinburgh EH9 3FB, UK. [3] SAFCell Inc. 36 S. Chester Ave, Pasadena, CA 91106, USA. Correspondence and requests for materials should be addressed to N.R. (email: n.radacsi@ed.ac.uk) or to K.P.G. (email: giapis@cheme.caltech.edu)

The promise of nanotechnology cannot be realised without inexpensive and efficient manufacturing of nanometer-sized objects, which must be further prevented from coalescing into larger objects devoid of nanoscale properties. This is particularly true for nanoparticles, which may possess special optical, electrical, magnetic, or catalytic properties[1,2] different from those of the bulk. Thus, supporting nanoparticles to keep them apart is also necessary to avail of their nanoscale properties over long periods of time. Lastly, immobilizing nanoparticles makes their handling easier and safer.

Incorporation of functional nanoparticles within an electrospun fibre is a timely topic in the field of electrospinning research[3]. Electrospinning is an inexpensive and versatile technique that uses high voltage for the production of fibres from polymers, inorganic materials, and composites, with diameters ranging from tens of nanometers to several micrometres[4]. Among several methods for producing nanofibres[5–12], the electrospinning technique has emerged as the most efficacious process for nanofibre production[13]. The process is scalable, can be tuned to high yields, and offers excellent control over nanofibre size and orientation[13]. Unfortunately, electrospinning from clear solutions (solutions in which the solute is fully dissolved) tends to produce nanofibres with smooth surface, which limits the surface area of as-deposited composites to being controlled only by fibre diameter. One way to bypass this limitation and increase total exposed surface area is to place nanoparticles on the nanofibre surface[14,15], a difficult task when the nanoparticles are pre-dispersed in the solution from which the fibres are formed. While a few may end up on the skin of the fibres, most nanoparticles become embedded into the fibres, and do not augment the surface area of the produced composite[16]. The benefits of increased surface area can be realized by a process that ensures facile formation and uniform distribution of particles on the fibre surface[17].

We report here the spontaneous formation of electrolyte salt nanoparticles on electrospun nanofibres containing polyvinylpyrrolidone (PVP) or polyvinyl alcohol (PVA), and a small amount of emeraldine base polyaniline (PANI). The electrolyte caesium dihydrogen phosphate (CDP) is an electrolyte used in intermediate-temperature solid acid fuel cells (SAFCs), both as a proton-conducting membrane and as an electrode ingredient. Fabrication of the cathode electrode is currently based on mechanically pressing conductive carbon paper together with some micron-sized CDP powder into porous discs, followed by chemical vapor deposition of a platinum-precursor to coat the exposed CDP surface with metallic platinum (Pt). The resulting structure ensures presence of a triple-phase boundary, where the electrolyte (CDP), the catalyst (Pt), and gas-phase (air) are in contact as needed for electrocatalysis. Maximizing triple phase boundary can boost performance substantially, which is the reason why the size and number of surface particles in contact with an electrically conductive and gas accessible Pt network is so important. Using smaller diameter CDP particles to form higher surface area porous electrodes runs into the particle-coalescence problem. Creating a CDP porous network by electrospray deposition of nanoparticle-decorated fibres provides a more resilient structure, which improves the power output of SAFCs[18]. Electrospun nanofibres can be covered with nanoparticles by immersing them into solutions of dispersed nanoparticles, by in situ reduction of an appropriate precursor at the nanofibre surface, or via hydrothermal processes[19–22]. These methods are relatively complex and very inefficient. Electrospray can also be combined with electrospinning to deposit nanoparticles directly on the surface of nanofibres[23,24]. However, the deposition is not homogeneous, and the nanoparticle density tends to be low, in line with the low electrospraying yields. The present work focuses on depositing fibres, loaded with large numbers of spontaneously formed CDP nanoparticles in a single step.

## Results

**Nanoparticle formation on nanofibres**. The process of spontaneous formation of pretzel-stick-like composite structures, consisting of nanoparticle-decorated nanofibres, by electrospinning is shown in Fig. 1. We produce these structures by nozzle-free electrospinning rather than conventional nozzle-based electrospinning using nozzles, in order to substantially enhance the yield of nanofibres significantly, as appropriate for industrial production. The process starts by preparing a clear solution of aqueous CDP mixed with polymer additives. The increased viscosity obtained by polymer addition is necessary for operating in the electrospinning mode, otherwise no fibres are formed. The solution is placed in a bath, into which a cylindrical metallic drum is partially immersed (Supplementary Fig. 1). Slow rotation of the drum forms a thin liquid film on its surface. Upon application of a sufficiently high voltage to the drum, the liquid film surface deforms. When electrostatic forces overcome the surface tension of the liquid solution, multiple Taylor cones form on the surface of the drum commencing the electrospinning process. The fibres ejected from the cones are deposited onto a biased (−5 kV DC) and heated substrate (120 °C). A heated counter-flow design is employed (Fig. 1), which produces a temperature gradient in the region of fibre whip instability and elongation, enabling electrospinning to take place at low polymer concentrations. As the nanofibres cure during and after deposition, nanoparticle crystals emerge spontaneously on their surface.

**Key process parameters**. Two parameters were identified to play a significant role in the process of forming nanoparticles on the fibre surface: solute concentration and type of polymer employed. The CDP salt and polymer concentrations were varied and their effect on nanostructure morphology and production rates was assessed. For the highest catalytic activity possible, the CDP-to-polymer ratio needs to be maximized, while preserving the triple phase boundary. This is because the polymer is not active in the oxygen reduction reaction but is necessary to enable the electrospinning process. Different CDP concentrations were mixed with various amounts of PVP or PVA (see Supplementary Table 1). PANI was added to all the solutions at fixed concentration (0.1 mg mL$^{-1}$) to increase the conductivity of the solid composite. The addition of PANI does not seem to impact nanoparticle formation (see Supplementary Fig. 2). When PVP was used in the electrospinning process, the nanoparticle formation was observed only when the CDP concentration was 50 mg mL$^{-1}$. Increasing PVP concentration at this fixed CDP value results in concomitant increase in fibre diameter, nanoparticle size, and nanoparticle density (Fig. 2). A sudden increase of PVP fibre diameter can be observed when transitioning from 35 to 40 mg mL$^{-1}$, which is attributed to the presence of the salt[25], the lower CDP content per unit volume and to larger solution viscosity[25–28]. Thus, by changing the PVP concentration, the size of both nanofibres and nanoparticles, and the spacing of such nanoparticles along the nanofibre surface can be controlled over a wide range. Using PVP at a concentration of 30 mg mL$^{-1}$, nanoparticle-decorated nanofibre composites can be achieved with a mean nanofibre diameter of 123.9 ± 32 nm, and a mean particle diameter of 104.5 ± 28 nm can be achieved (Supplementary Fig. 3). At this PVP concentration, the nanoparticle density was 29 particles μm$^{-2}$.

The CDP particles are present not only on the surface of the fibres, but also in their interior, as observed by cross-sectional scanning electron microscopy of CDP-PVP-PANI fibres, with images shown in Fig. 3a and Supplementary Fig. 4. CDP particles

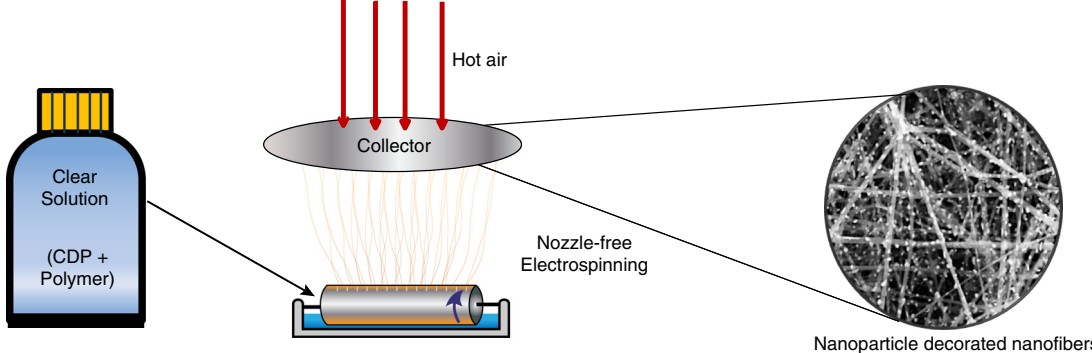

**Fig. 1** Schematic illustration of the electrospinning process. The nanoparticle-decorated nanofibres are fabricated from a clear solution in a single step, containing dissolved caesium dihydrogen phosphate (CDP) and polymers. Multiple Taylor cones form on the rotating electrode immersed into the solution. Hot air is blown on the collector electrode, enabling electrospinning to take place at low polymer concentrations. Uniform fibres with CDP nanoparticles are deposited over a large area

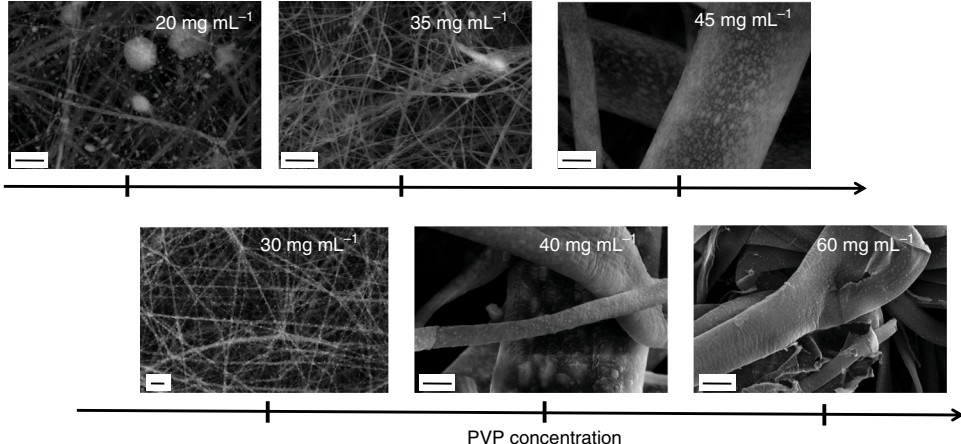

**Fig. 2** Scanning electron microscope images of as-spun composite samples. The images show the nanoparticle-decorated caesium dihydrogen phosphate (CDP)-polyvinylpyrrolidone (PVP)-polyaniline (PANI) samples as a function of PVP concentration. Polymer beads are not present at PVP concentrations above 30 mg mL$^{-1}$. The size and density of nanoparticles on the fibres increase with increasing PVP concentration. The mean fibre diameter is the smallest with a PVP concentration 30 mg mL$^{-1}$. Scale bars represent 1 μm

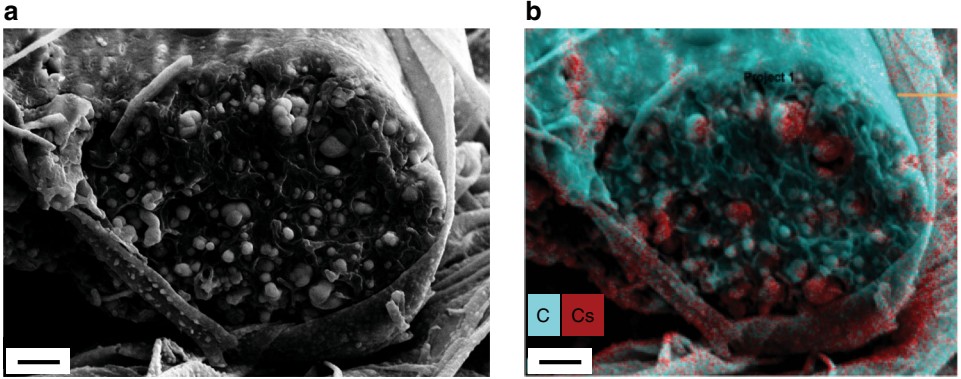

**Fig. 3** Cross-section scanning electron microscope (SEM) image. The image shows a large electrospun caesium dihydrogen phosphate (CDP)-polyvinylpyrrolidone (PVP)-polyaniline (PANI) fibre with CDP particles inside the fibre and on the surface. **a** SEM image showing CDP particles ranging from 190 nm to 1.2 μm inside the electrospun fibre. **b** Energy dispersive X-Ray spectroscopy map of the same cross-section, showing the caesium (Cs) in red, and the carbon (C) in turquoise. Scale bars are 2 μm

with different sizes (from 190 nm to 1.2 μm, with an average of 440 nm) can be observed, many of them placed inside a polymer fibre pocket. Energy dispersive X-ray spectroscopy (EDS) analysis reveals that the nanoparticles indeed contain caesium (Cs) atoms (Fig. 3b). It also appears that the fibre surface is coated. This coating seems to be polymer (PVP), which likely prevents CDP nanoparticles from participating in the oxygen reduction reaction at the triple phase boundary and, thus, it should be reduced.

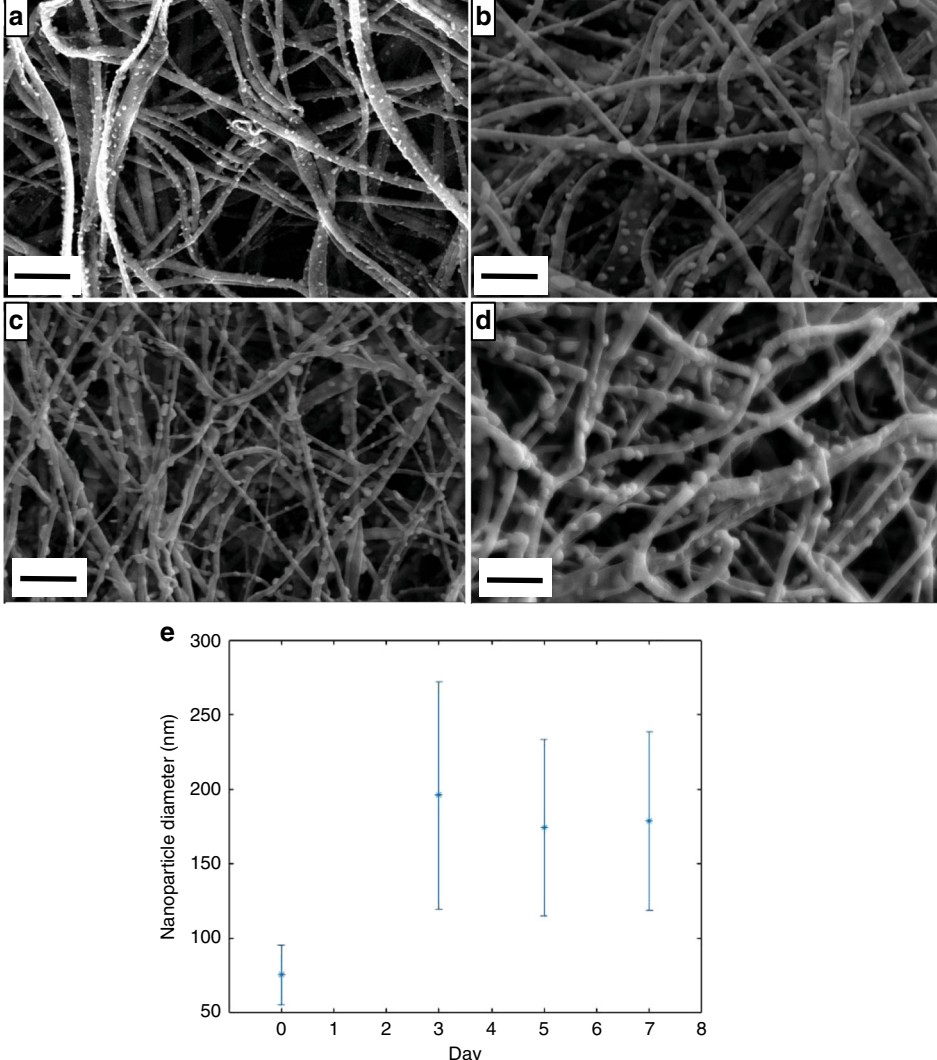

**Fig. 4** Effects of constant relative humidity on the electrospun samples over time. The caesium dihydrogen phosphate-polyvinylpyrrolidone-polyaniline (CDP-PVP-PANI) composites show changes in the nanoparticle diameter over time during constant temperature and humidity (23 °C and 35% relative humidity). **a** As-spun sample. **b** Day 3. **c** Day 5. **d** Day 7. Scale bars represent 1 μm. **e** Plot showing the size and size distribution evolution over time. Error bars show standard deviation ($n = 106$). Source data are provided as a Source Data file

The electrospun CDP-PVP-PANI composite material was exposed to $35 \pm 2\%$ humidity over a course of 7 days at 23 °C, and scanning electron microscopy (SEM) images were taken on day 0, day 3, day 5 and day 7. The results show that the nanoparticles decorating the PVP fibres grew from 75 nm to 190 nm between day 0 and day 3, and then shrank to ~175 nm (Fig. 4). The humidity effect on the nanoparticle size was investigated on electrospun CDP-PVP-PANI material (with nanoparticles on the fibre surface), which was cut into three pieces to be studied under different humidity conditions. The samples were stored for seven days under 0%, 55% and 73% relative humidity (RH). When the samples were kept in vacuum (0% RH), the nanostructure was preserved. When the samples were exposed to 55% or 73% RH, the nanofibres disappeared, and a mat with the nanoparticles was formed (Supplementary Fig. 5).

The effect of the polymer type on the nanoparticle formation was considered next. Using PVA instead of PVP resulted in nanoparticle formation on the electrospun fibre surface only when the CDP concentration was 30 mg mL$^{-1}$. The polymer type affects the nanoparticle size and density, albeit at different concentrations. The nanofibre composite mat produced from the 15 mg mL$^{-1}$ concentration solution featured nanoparticle size of $188.5 \pm 43$ nm, slightly lower than the one fabricated from the 30 mg mL$^{-1}$ PVA solution, which was $218 \pm 69$ nm (Fig. 5). The nanoparticle density was higher for the lower PVA concentration (15 mg mL$^{-1}$), than for the higher PVA concentration (30 mg mL$^{-1}$): 28 vs. 17 particles μm$^{-2}$, respectively. The nanoparticle density as a function of polymer concentration shows opposite trends for PVA, as compared to PVP. This is likely due to the different nature of the two polymers.

The formation of the multiple Taylor cones on the drum electrode surface (Fig. 6a) results in rapid deposition of nanofibres over large areas. An example of a 109 cm$^2$ area CDP-PVP-PANI composite SAFC electrode, with ~100 nm electrolyte nanoparticles on the surface on the 50–200 nm nanofibres is shown in Fig. 6b (see Supplementary Fig. 3 for SEM images and histograms showing the size distribution). The deposition rate can be up to 900 mg h$^{-1}$ in the mentioned laboratory scale setup[29].

The production rate of the composite nanofibres depends on the solution composition, applied potential difference, substrate temperature and processing time. Figure 7 shows the linear increase of the weight of deposited material on a 2.3-inch round

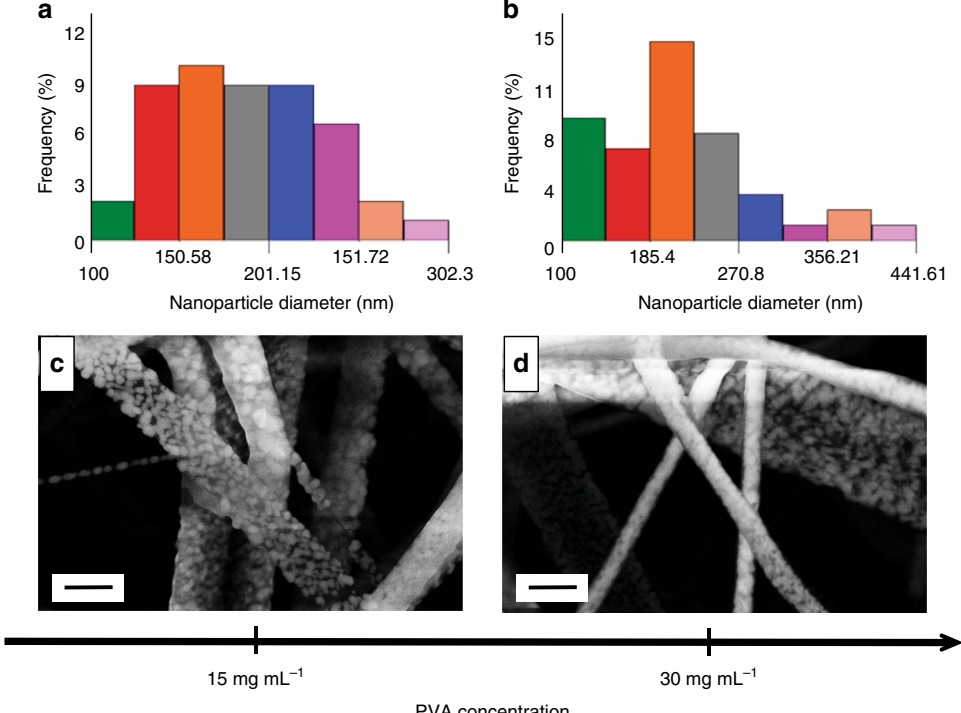

**Fig. 5** Scanning electron microscope images with corresponding histograms. The images show the as-spun nanoparticle-decorated caesium dihydrogen phosphate (CDP)-polyvinyl alcohol (PVA)-polyaniline (PANI) samples as a function of PVA concentration. **a** Histogram showing the size distribution for the sample from 15 mg mL$^{-1}$ PVA concentration. The mean nanoparticle size is 188.5 ± 43 nm. **b** Histogram showing the size distribution for the sample from the 30 mg mL$^{-1}$ PVA solution. The mean nanoparticle size is 218 ± 69 nm. **c** Scanning electron microscopy (SEM) image of the sample from 15 mg mL$^{-1}$ PVA solution. The nanoparticle density is 28 particles µm$^{-2}$. **d** Scanning electron microscope image of the sample produced from the 30 mg mL$^{-1}$ PVA solution. Nanoparticle density: 17 particles µm$^{-2}$. Scale bars are 1 µm

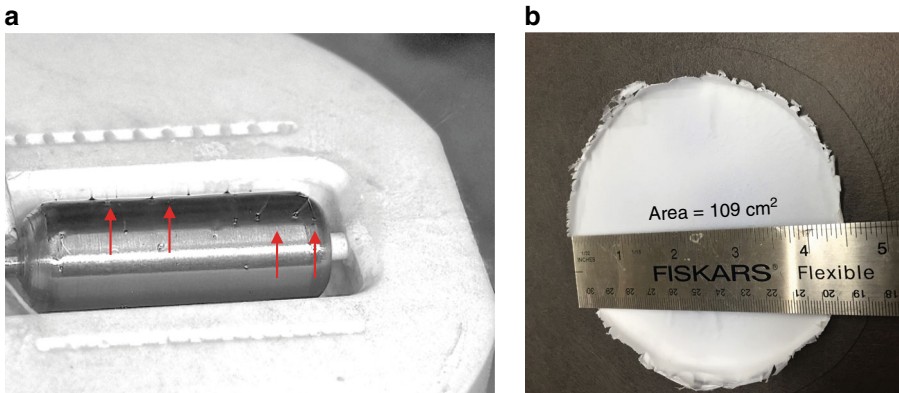

**Fig. 6** Image of the rotating electrode and the collected sample on a carbon paper. Needle-free electrospinning can produce homogeneous and uniform films over a large area. **a** Drum cylinder with multiple Taylor-cones (some are pointed at by red arrows) emerging on the surface of the rotating stainless-steel electrode at an applied potential of 50 kV. **b** Caesium dihydrogen phosphate-polyvinylpyrrolidone-polyaniline (CDP-PVP-PANI) composite material electrospun directly on a carbon paper from a clear solution. The film was obtained within 10 minutes of electrospinning deposition

flat collector substrate over processing time. The electrospinning production rate is approximately 10 times that of electrospraying. The solution used in this study was CDP:PVP:PANI of 50:30:0.1 mg mL$^{-1}$ (PANI is used to increase the conductivity of the collected sample, as PANI is an electrically conductive polymer in the emeraldine base form[30]). During the electrospinning process, the current measured on the collector electrode was stable 111 ±2 µA (Supplementary Fig. 6). The addition of the 10 vol% dimethylformamide (DMF) increases the production yield significantly. A solution of CDP:PVA:PANI=30:30:0.1 mg mL$^{-1}$ yields material at 182 mg h$^{-1}$, which drops to 54 mg h$^{-1}$ when no

PANI-DMF solution was added. The presence of the PANI in the solution did not affect the production rate of the sample in the first 30 minutes of electrospinning.

**Electrochemical performance**. The reduction of the polymer was performed by treating the samples at 300 °C in a furnace, though only part of the polymer was removed at that temperature (see Supplementary Fig. 7). This step is required to improve electrochemical performance, because PVP and PVA are not active in the oxygen reduction reaction. However, if too much polymer is

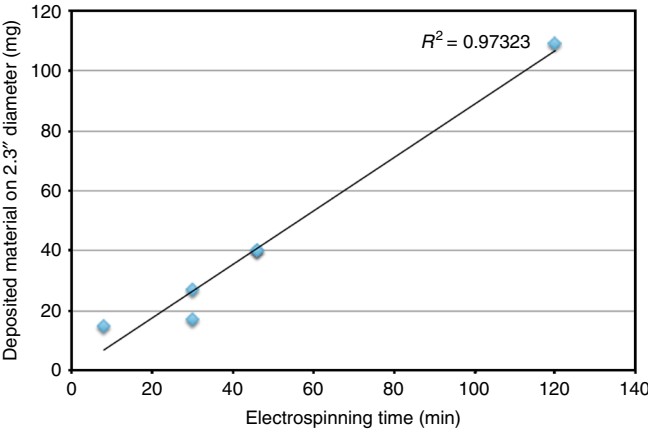

**Fig. 7** The production rate plotted over time. The weight of deposited material on a 2.3″ diameter collector shows a linear increase over time

removed, the nanostructure collapses, leading to poor fuel cell performance. After extensive experimentation, it was concluded that 300 °C was the optimal heat treatment temperature. As the heat treatment causes the electrospun sample to peel off the carbon paper substrate, material collected with a rotating drum was used for surface area and electrochemical characterisations. Brunauer-Emmett-Teller (BET) surface area analysis of as-spun CDP-PVP-PANI composite fibres showed a surface area of 21 m$^2$ g$^{-1}$, which is more than 3 times higher than the sample electrospun from a solution without CDP (3.2 m$^2$ g$^{-1}$), and almost 9-fold higher when compared to standard cathode electrodes (2.4 m$^2$ g$^{-1}$). Electrochemical impedance and fuel cell testing were performed on the CDP:PVP:PANI sample (50:30:0.1 mg mL$^{-1}$) after metal-organic chemical vapor deposition (MOCVD) of Pt, which increased the sample weight by 32% (see Supplementary Note 1 and Supplementary Fig. 8). X-ray diffraction (XRD) confirmed that the CDP was in monoclinic state after the heat treatment, and the MOCVD process (Supplementary Fig. 9). At the 2″ diameter benchmark (~15 cm$^2$ active area), the fabricated nanocomposite cathode possessed area-specific resistance value of 20 mΩ cm$^2$ (Supplementary Note 2) as shown by the area-specific resistance measurement (Supplementary Fig. 10). Using electrospun nanoparticle-decorated fibres for the cathode improved the cell voltage at every current density as compared to the state-of-the-art standard electrode (with 2.89 mg cm$^{-2}$ Pt catalyst deposited on 50 μm layer electrode of 800 nm pure CDP particles, pressed together). The significant improvement in performance, shown in Fig. 8a, is attributed to the increase in available surface area for electrocatalysis at the triple-phase boundary. Additional measurements showed that the electrospun 2″ fuel cell had similar peak power density and stability as the standard electrode, with a maximum of ~260 mW cm$^{-2}$ (Fig. 8b).

Electrospun samples with PVA did not yield electrochemical performance comparable to those with PVP. SEM imaging of samples with PVA showed that the nanostructure was lost after heat treatment at 300 °C (Supplementary Fig. 11). Thus, the samples deposited with PVA exhibited deficient electrochemical performance in comparison to using PVP (Supplementary Fig. 12).

## Discussion
Spontaneous nanoparticle formation on the fibre surface appears to require a crystallisable salt at a certain concentration. The nanoparticle formation mechanism appears to require that solute is drawn out of the nanofibres. Once nucleation occurs, the

crystals grow by continued solute transport through the fibre skin. This may occur by dissolution of the solute into nanopores of the polymeric nanofibre surface, assisted by ambient humidity or any remaining solvent (water) in the fibre as a result of incomplete drying (see Supplementary Fig. 14). Different solvents with different vapor pressures affect significantly the nanoparticle formation mechanism. CDP, PVP and PVA are all soluble in water[31,32], but the solubility of CDP in water is much higher (around 1300 mg mL$^{-1}$)[31] than the solubility of PVP and PVA in water (100 mg mL$^{-1}$ at room temperature for PVP[33], and 110 mg mL$^{-1}$ at 85 °C for PVA[34]). During the electrospinning process, the solution mixture becomes supersaturated with the polymer (PVP or PVA) much earlier than that condition is reached for CDP, and the polymer precipitates first. That is, CDP is still in a dissolved state when the polymer fibre solidifies. Methanol has significantly higher vapor pressure than the other two solvents that were used, water and DMF[35–37], thus it will evaporate first from the mixture. Water has a higher vapor pressure than DMF[36,37], thus it evaporates faster than DMF during the electrospinning process. DMF does not dissolve CDP, so it acts as an anti-solvent in the water-DMF mixture, promoting CDP crystallisation during the process[38]. Once supersaturation conditions are reached, the driving force for nucleation from the liquid solution pockets inside the fibre takes over, forming nanoparticles in the available confined space[39] (e.g., via heterogeneous nucleation). As the nanofibre containing the solute dries and cures, out-diffusion supplies additional solute to grow the nanoparticles from inside the fibres (see Fig. 3. and Supplementary Fig. 3).

We surmise that no nanoparticles are detected below a critical salt concentration because of their small size. Ostwald ripening[40] likely prevents nanoparticle formation at larger salt concentrations by limiting salt out-diffusion. The elevated temperature and the needle-free electrospinning process do not seem to play a role in nanoparticle formation. Indeed, nanofibres laden with nanoparticles (similar to Fig. 2) were observed with conventional (needle-based) electrospinning at ambient temperature when using high (80 mg mL$^{-1}$) PVP concentration (see Supplementary Note 3 and Supplementary Fig. 13). The heated air counter-flow design (Fig. 1 and Supplementary Fig. 1) is needed to enable the occurrence of electrospinning at low polymer concentrations (<40 mg mL$^{-1}$), where otherwise no electrospinning is observed. The low PVP/PVA concentration is desired for direct electrospinning of fuel cell electrodes, as these polymers are not active in electrocatalysis.

The linear increase in the sample production rate is likely due to the added electrically conductive polymer, the emeraldine base PANI[30], as PVP and CDP are poor electrical conductors[41,42]. Nanoparticle-decorated nanofibres formed with PVP outperformed standard electrodes that are made of pressed 800-nm CDP particles and employed in state-of-the-art fuel cells, likely due to larger surface area and triple-phase boundary. In contrast, fuel cells with electrodes made from nanoparticle-decorated nanofibres formed with PVA have inferior performance, attributed to the different thermal decomposition profile of PVA versus PVP[43]. For most applications, it is desirable to support the nanoparticles with a polymer backbone to prevent loss of the high surface area, see Supplementary Fig. 10. Nanoparticle-decorated nanofibres formed by electrospinning offer such a backbone and thus have potential in other application areas, where such structures are important.

## Methods
**Solution preparation**. CsH$_2$PO$_4$ (CDP) was prepared by precipitation from an aqueous solution of Cs$_2$CO$_3$ and H$_3$PO4, as described previously[44]. Cs$_2$CO$_3$ (99.9%), purchased from Alfa Aesar, and H$_3$PO$_4$ (ACS, 85% w/w aqueous solution) were combined in a molar ratio of 1:2 in aqueous solution and subsequently

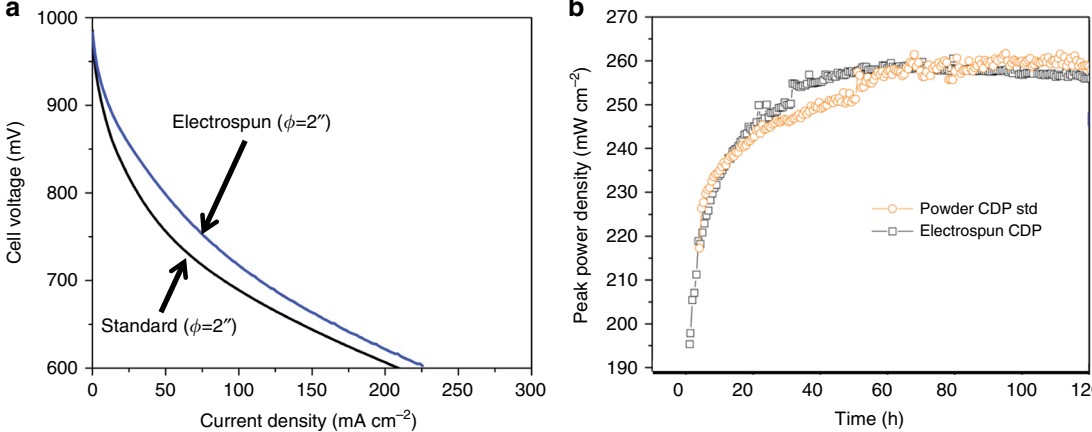

**Fig. 8** Electrochemical performance of the nanoparticle-decorated sample. The sample material is caesium dihydrogen phosphate (CDP)-polyvinylpyrrolidone (PVP)-polyaniline (PANI) composite, used in a 2″ diameter (∅) fuel cell. **a** The cell voltage at different current densities. The blue line represents the electrospun sample with the nanoparticle-decorated nanofibres. The black line shows SAFCell Inc.'s standard (std) electrode. The electrospun sample has improved cell voltage at every current density. **b** The peak power density of the electrospun CDP sample with PVP-PANI is comparable with the standard powder SAFCell Inc. fuel cell electrode. The electrospun fuel cell power does not decrease in the first 60 hours of operation

precipitated in methanol, followed by drying of the collected solid in air at 120 °C. The synthesis of pure CDP was confirmed by X-ray powder diffraction. Poly-vinylpyrrolidone (PVP; $M_w$ 1,300,000 u) and polyvinyl alcohol (PVA; $M_w$ 85,000–124,000 u) were purchased from Alfa Aesar. Polyaniline (PANI; emeraldine base, average $M_w$ 50,000 u) and anhydrous dimethylformamide (DMF) were purchased from Sigma-Aldrich. The 99.8% methanol was purchased from BDH. All chemicals were used without further purification. Aqueous solutions were prepared as follows: For the PVP solutions, 5 g CDP and 3-6 g PVP were dissolved in 50 mL deionized (DI) water and 40 mL methanol in separate bottles, respectively. In a third beaker, 10 mg PANI was dissolved in 10 mL DMF, making a concentration of 0.1 mg mL$^{-1}$. When all the three beakers showed clear solutions, they were mixed together. For the PVA solution preparation, 3 g CDP was dissolved first in 50 mL DI water, then 1.5 or 3 g solid PVA was added to the solution. Upon observing a clear solution, 40 mL methanol was added to it. 10 mg PANI was dissolved in 10 mL DMF in a different beaker, and was added to the main solution.

**Electrospinning setup and procedure**. A home-built nozzle-free electrospinning setup was used to produce the nanoparticle-decorated nanofibres (Supplementary Fig. 1). The potential difference for the drum electrode was provided by a Spellman SL2000 +60 kV power supply. An Urbest 5 r.p.m. high torque permanent magnetic DC gear motor was used to rotate the stainless-steel electrode in the solution bath that was connected with a 162 mm long, 6 mm diameter PTFE rod. A −30 kV Glassman ER30N10 power supply was used for biasing the collector electrode.

During the fabrication of the nanoparticle-decorated nanofibres with PVP and PANI, the CDP concentration was fixed at 50 mg mL$^{-1}$, the potential difference between the electrodes was 50 kV (+45 kV DC on the rotating electrode immersed in the solution, and −5 kV DC on the collector electrode), and the distance between the top of the rotating electrode and collector was 140 mm. Two different types of collector were used: a fixed 2 mm thick copper plate with a diameter of 130 mm, and a rotating stainless-steel electrode, which was 80 mm long and 20 mm in diameter (Supplementary Fig. 1). The fix copper plate collector had the advantage of depositing nanofibres directly on carbon paper, while the rotating collector gave the possibility of collecting thick samples. Toray carbon paper without any PTFE coating (purchased from Fuel Cell Earth, USA) was used with the fixed collector plate, mounting it below the copper plate, securing it with a help of a PTFE ring. The current on the collector was measured during the electrospinning process by a digital multimeter (Cen-Tech, China) that was connected in series with the negative power supply. A 1600 W heat gun (Chicago Electric, USA) directed on the substrate provided hot air, which heated the substrate to 120 °C. The distance between the heat gun and collector electrode was 192 mm. The rotating collector electrode was revolving with 300 rpm, while the rotating electrode immersed in the solution bath was rotating with 3 rpm. A sharp rectangular carbon blade was used to remove the electrospun sample from the rotating collector electrode. The reproducibility of the CDP nanoparticle-decorated nanofibres was investigated in a commercial needle-based electrospinning apparatus (NovaSpider, CIC nanoGUNE, Spain). The electrospun composite mat was treated in a Barnsted Thermolyne Series 48000 box furnace to remove the remaining solvents and part of the polymer (Supplementary Note 1). The temperature was increased in 3 steps to 300 °C (200 °C for 2 h, then 230 °C for 2 h, and finally 300 °C for 12 h) using a heating rate of 5 °C min$^{-1}$. The first 2 steps are necessary to dehydrate the CDP to the thermodynamically stable CsPO$_3$[45], so that

the electrolyte stays solid at 300 °C, instead of becoming liquid. Then, Pt nanoparticles were deposited on the electrodes by chemical vapor deposition, using a StableTemp 281A vacuum oven at 210 °C and 0.3 bar, following strictly the procedure described by Papandrew et al[46]. As the Pt loading in our study was 29 wt %, a uniform deposition of 3.7 nm Pt nanoparticles was achieved in the MOCVD process[46]. Next, the heat-treated product with the Pt nanoparticle deposition was sieved, and hand-spread on a 2″ electrode for fuel cell testing (see Supplementary Note 1).

**Physical and electrochemical characterisation**. For visualisation of the nano-particles on the fibres, a ZEISS 1550 field emission scanning electron microscopy (SEM) was used at an accelerating voltage of 10 kV. For imaging the sample cross-sections, a Zeiss Crossbeam 550 focused ion beam SEM was used at an accelerating voltage of 2–3 kV. The diameters of the nanoparticles and nanofibres were mea-sured with the image processing program, ImageJ.

The composite samples prepared by the needleless electrospinning technique were characterised by XRD, using a PANalytical X'Pert Pro instrument and a Bruker D2. On both instruments, the position was up to 70° 2theta, and the step size was 0.05°.

For the electrochemical characterisation, the two half-cells were pressed together with 1 tons for 3 s. Two stainless steel porous discs were placed on each side of the symmetric cell in order to give uniform gas diffusion to the surface of the electrode. Electrical data were collected under humidified hydrogen using an impedance analyzer (Eco Chemie Autolab PGSTAT302 equipped with a frequency response analysis module) and operating at a voltage amplitude of 10 mV over frequencies ranging from 10 to 1 MHz. Hydrogen was supplied at a rate of 70 sccm, and humidified by flowing through a water bubbler held at 80 °C. The operating conditions for the fuel cell measurements were: 0.30 atm pH$_2$O, 250 °C; flow rates of 70 sccm 100% H$_2$ at the anode and 150 sccm air at the cathode.

## Data availability
The data that support the findings of this study are available from the corre-sponding author upon reasonable request. The source data underlying Fig. 4e are provided as a Source Data file.

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

## Acknowledgements

The authors thank Miguel Gonzalez of Caltech and Michel Vong of The University of Edinburgh for assistance with the project. Thomas Glen of The University of Edinburgh helped with the focused ion beam scanning electron microscopy. The information, data, or work presented herein was funded in part by the Advanced Research Projects Agency-Energy (ARPA-E), U.S. Department of Energy, under Award Number DE-AR0000495.

## Author contributions

N.R. and K.P.G. designed the experiment and co-wrote the paper. N.R. performed the experiments and characterised the electrospun materials, F.D.C. built the fuel cells, and tested the samples. C.R.I.C. interpreted the electrochemical data. K.P.G. supervised the project.

## Additional information

**Competing interests:** The authors declare no competing interests.

