## [Peer Review File · Nature Communications]

Reviewers' comments:

Reviewer #1 (Remarks to the Author):

Comments:

The authors fabricated CDP/PVP composite fibers by nozzle-free electrospinning for fuel cell electrodes. In this paper, the authors emphasized the high production speed of this needleless electrospinning technique, the high surface area and the good performance for SAFC. However, this manuscript was not well prepared and many issues have not been explained clearly. In addition, the method is not new at all. The reviewer does not believe that the novelty is enough for publishing in this journal.

1. Some important data in SI should be added to the main text, such as the fiber production speeds of the traditional electrospinning with one needle and the nozzle-free electrospinning, the power density of the CDP based and commercial SAFCs, so that it can give the readers a more intuitive understanding of the high production speed, not just written in words.
2. CDP NPs can be spontaneously formed on the fiber surfaces and further increasing the surface area, do CDP NPs exist within the fibers? Compared with pure PVP fibers, how much has the surface area increased?
3. For the discussion part, the authors claimed that DMF promotes CDP crystallization, however, the mechanism needs to be proved. What's the function of PAN that added in the electrospun solution?
4. For the preparation of the fuel cell electrode, why choose 300 °C as the final temperature? The temperature was increased in 3 steps (first 200 °C for 2 h, then 230 °C for 2 h, and finally 300 °C), what's the effect of the first two steps?
5. Some statements are not rigorous. For example, about the electrospun solution, there is also PAN in the fiber, however, there is no description about PAN in the main text, just appeared in the experiment section; There are several different abbreviations of solid acid fuel cells; The authors think NPs on the fiber surface could increase the surface area, there is no data or reference to confirm it; Some TEM images are not clear.

Reviewer #2 (Remarks to the Author):

1. The topic of the manuscript does not reflect the actual content, therefore, it is suggested to revise the topic accordingly. Also, the as-synthesized CDP nanoparticles decorated on nanofiber is used as an electrolyte in the fuel cell, not for fuel cell electrodes.
2. Please revise the abstract by describing with more informative and specific findings of the study.
3. line42-44; please discuss how to address the issue as mentioned in the introduction through literature reviews.
4. line 58-61; please justify what is the approach attempted in this study.
5. line 71; it would be better if the author can add in a schematic illustration of the nozzle-free electrospinning process set up along with Fig. S1.
6. line 84-87; please tabulated the results (e.g. CDP size, nanofiber diameter, nanoparticles density etc) of the samples prepared via varied parameters.
7. Fig 2; the diameter of the nanofiber increased significantly at PVP 40 mg/mL. Please discuss.
8. line 103; please explain what is the role of PVA and PVP in the electrospinning process.

9. line 103-104; how to ensure there is no CDP at other concentration in the nanofiber? please clarify if there is CDP embedded into the fibers?

10. line 127-130; the written text "The electrospun composite..." is more appropriate to be removed to the Methods section.

11. line 137-138. the author mentioned the cell voltage is increased due to the triple phase boundary. this is not justifiable based on the test performed in this study.

12. The discussion section needs to be revised significantly before it can be further for publication. The depth of discussion and comparison with state-of-the art literature data is absent, making it difficult to determine whether significant advances have been made, or new insights gained.

13. line 208; the tube furnace mentioned does not look alike with the Fig. S4, which is a box furnace. please check.

14. line 212; typo of "Fig. SX"

15. line 213; there is not Fig. S8 in the appendix.

15. Please add in the method of Pt deposition in the Methods section.

Reviewer #3 (Remarks to the Author):

Review for manuscript: NCOMMS-18-03038

Title: Spontaneous formation of nanoparticles on electrospun fibers for fuel cell electrodes

This manuscript describes the fabrication of electrospun nanofibers by nozzle-free electrospinning and the spontaneous formation of cesium dihydrogen phosphate (CDP) nanoparticles onto their surfaces under specific experimental conditions. The generation of the CDP nanoparticles resulted to an enhanced electrocatalytic performance of the heat-treated membrane decorated with Pt catalytic nanoparticles, owned to the increased surface area of the electrospun CDP/polymer nanocomposite fibers.

The presented work is interesting. However, there are some points that should be addressed by the authors before considering publication in Nature Communications.

Major comments

- During the electrospinning process, the produced fibers were deposited on a substrate that was heated at 120 °C. Does the temperature of the heated substrate plays a key role in the spontaneous formation mechanism of the CDP nanoparticles? Would it be possible to control/tune the CDP nanoparticle diameters by varying this temperature?
- Why in the case of PVP, upon increasing the PVP concentration the nanoparticle density increases (lines 89, 90), while in the case of PVA the opposite is observed? (see lines 108, 109 – where it is stated that the nanoparticle density was higher for the lower PVA concentration).
- Additional experimental data (i.e. SEM images) should be provided for the CDP-decorated fibers (a) after heat treatment at 300 °C for 12 hrs and (b) after Pt deposition. Does the heating step affect the CDP sizes and size distribution? What is the average size and size distribution of the deposited Pt nanoparticles?
- In Figure 5, do the 2 graphs derive from single measurements? If not, error bars must be

included. Is there a statistically significant difference between the 2 plots? Is there any information related to the electrocatalytic performance of the PVA-CDP analogues?

- In the discussion part, the authors should give more experimental evidence to support their statements related to the spontaneous formation mechanism of the CDP nanoparticles on the fibers' surfaces. Why this spontaneous process can only occur at a specific salt concentration? Is there any evidence of the existence of nanopores onto the fibers' surfaces that – according to the authors – may be involved in the CDP formation mechanism?

- In Figure S2, do the peaks observed for the CDP nanoparticle crystals correspond to monoclinic CsH_2PO_4 ? In the XRD spectrum provided for the CDP nanocrystals I would also suggest to include the XRD spectrum of Pt-deposited analogue.

Minor comments

- In the introduction part, (page 3, lines 58-62), the authors mention several ways employed for the anchoring of nanoparticles onto the surfaces of electrospun fibers, including their immersion in solutions of dispersed nanoparticles, the in situ reduction of an appropriate precursor at their surfaces or the use of hydrothermal processes. Electro spraying is another way of depositing nanoparticles onto the surfaces of electrospun fibers and it can be performed simultaneously with electrospinning therefore avoiding time-consuming and multi-step procedures...A few references related to the use of electro spraying in the deposition of nanoparticles onto the surfaces of electrospun fibers should be included.

- In Figure 2, the presence of CDP nanoparticles is not obvious in the case of the 60 mg/mL PVP concentration.

- Line 172. Please provide the solubility of the polymers in water.

- Line 212. Figure SX. Please provide the correct Figure numbering.

- Line 212. Figure S8 – there is no Figure S8.

- "Electrospray deposition" should be removed from keywords.

Reviewer #1 (Remarks to the Author):

Comments:

The authors fabricated CDP/PVP composite fibers by nozzle-free electrospinning for fuel cell electrodes. In this paper, the authors emphasized the high production speed of this needleless electrospinning technique, the high surface area and the good performance for SAFC. However, this manuscript was not well prepared and many issues have not been explained clearly. In addition, the method is not new at all. The reviewer does not believe that the novelty is enough for publishing in this journal.

A: Thank you very much for your comments, which made us think more deeply about presenting the novelty of the findings. The manuscript has been significantly improved by addressing all of your excellent comments. The quality of the manuscript has been enhanced, and the results have been clarified. We believe that our results are novel, as there was no publication or patent found describing similar findings. After several discussions with patent lawyers, we have filed a patent application (U.S. Patent Application No. 15/623220 (14 June, 2017)). The reported method for fabricating nanoparticle decorated nanofibers in single-step electrospinning process shows promising potential for energy, catalysis or sensing applications, and can be produced at large scales. Please find below the response to your comments with green color.

1. Some important data in SI should be added to the main text, such as the fiber production speeds of the traditional electrospinning with one needle and the nozzle-free electrospinning, the power density of the CDP based and commercial SAFCs, so that it can give the readers a more intuitive understanding of the high production speed, not just written in words.

A: All the suggested data has been moved from the SI to the main body.

2. CDP NPs can be spontaneously formed on the fiber surfaces and further increasing the surface area, do CDP NPs exist within the fibers? Compared with pure PVP fibers, how much has the surface area increased?

A: Yes, we believe that the nanoparticles (NPs) can be spontaneously formed on the fiber surfaces in the described process. Our proposed mechanism, and the SEM images taken right after electrospinning suggest this. The CDP NPs can exist within the fibers, as shown by SEM image taken of the cross-section of these fibers. The fibers were cut by FIB. All of these findings have been added to the manuscript.

When there was no CDP in the electrospinning solution, that was otherwise the same as used in the study, the surface area of the sample was 3.2 g/m²

(while with CDP the surface area was 21 g/m²). This new result has been added to the results section of the manuscript.

3. For the discussion part, the authors claimed that DMF promotes CDP crystallization, however, the mechanism needs to be proved. What's the function of PAN that added in the electrospun solution?

A: DMF promotes crystallization of CDP in this case, as it is an anti-solvent for CDP in the used mixture. We investigated this phenomena in the laboratory, and indeed CDP crystallizes when excess DMF is added to the CDP solution. This has been added to the manuscript along with a reference for anti-solvent crystallization. The function of the PANI (emeraldine base) polymer is to increase the sample conductivity, which has been added to the manuscript.

4. For the preparation of the fuel cell electrode, why choose 300 °C as the final temperature ? The temperature was increased in 3 steps (first 200 °C for 2 h, then 230 °C for 2 h, and finally 300 °C), what's the effect of the first two steps ?

A: For the preparation of the fuel cell electrode, 300 °C temperature was chosen as at that temperature around half of the polymer content was reduced. The polymer content of in samples had to be reduced for good electrochemical performance, as PVP and PVA are not active in the oxygen reduction reaction. However, if too much polymer is removed, the nanostructure diminishes, leading to poor fuel cell performance. After several experiments were performed to investigate the heat effect at different temperatures (250°C, 300°C and 350 °C) during the furnace treatment of the electrospun samples, it was concluded that 300 °C was the optimal temperature to remove part of the polymer, while leaving a backbone for the CDP nanoparticles.

The first 2 steps are necessary to dehydrate the CDP to the thermodynamically stable CsPO₃, so that the electrolyte stays solid at 300 °C , instead of becoming liquid, as discussed by Taninouichi et al.

All of this information along with the references has been added to the manuscript.

5. Some statements are not rigorous. For example, about the electrospun solution, there is also PAN in the fiber, however, there is no description about PAN in the main text, just appeared in the experiment section; There are several different abbreviations of solid acid fuel cells; The authors think NPs on the fiber surface could increase the surface area, there is no data or reference to confirm it; Some TEM images are not clear.

A: This is a good point that lead to rewriting the paper making sure that all the

details are clear to the reader. The polymer, polyaniline (PANI), has been added to the text with explaining its role, and the abbreviations have been corrected. The surface area with and without the nanoparticles on the surface have been added to the main text body. The microscope images have been corrected, and many new images and illustrations have been added with detailed explanations.

Reviewer #2 (Remarks to the Author):

1. The topic of the manuscript does not reflect the actual content, therefore, it is suggested to revise the topic accordingly. Also, the as-synthesized CDP nanoparticles decorated on nanofiber is used as an electrolyte in the fuel cell, not for fuel cell electrodes.

A: Thank you for your observation. The manuscript has been substantially revised to reflect the actual content. The as-synthesized CDP nanoparticles are indeed electrolyte particles, but they can be used in the cathode assembly. Industry makes the cathode by pressing conductive carbon paper together with a small amount of micron-sized CDP powder into a porous pellet, making a $\sim 50 \mu\text{m}$ film. This is followed up by chemical vapor deposition of a platinum-precursor to coat the exposed CDP particle surface with platinum (Pt). The resulting cathode structure provides a good triple phase boundary needed for catalysis. At this boundary the electrolyte particles (CDP), catalyst particles (Pt) and the gas phase (oxygen) are in contact. Indeed, CDP is electrolyte, but here CDP nanoparticles are used in the cathode assembly, to increase the surface area of this triple phase boundary. So the porous CDP nanostructure on the cathode is for increasing the oxygen reduction reaction at the triple phase boundary. This has been added to the manuscript to clarify why CDP is part of the cathode assembly.

2. Please revise the abstract by describing with more informative and specific findings of the study.

A: The abstract has been revised and corrected by adding more informative and specific findings of the study.

3. line42-44; please discuss how to address the issue as mentioned in the introduction through literature reviews.

A: Some solutions to the issue has been added by citing other recent papers.

4. line 58-61; please justify what is the approach attempted in this study.

A: Thank you for your comment. The following sentence has been added to make the presented approach clear: “In this study, a novel single-step electrospinning approach is presented for decorating electrospun nanofibers with nanoparticles.”

5. line 71; it would be better if the author can add in a schematic illustration of the nozzle-free electrospinning process set up along with Fig. S1.

A: A schematic illustration of the nozzle-free electrospinning setup has been added to Fig. 1, and two additional schematic illustrations have been added to Supplementary Fig. 1.

6. line 84-87; please tabulated the results (e.g. CDP size, nanofiber diameter, nanoparticles density etc) of the samples prepared via varied parameters.

A: Thank you for the suggestion. A table has been added to the supplementary information as Supplementary Table 1.

7. Fig 2; the diameter of the nanofiber increased significantly at PVP 40 mg/mL. Please discuss.

A: Thank you for the comment. It is known that addition of salts can decrease the diameter of electrospun fibers, and a reference about this has been added to the manuscript. Since the CDP concentration is fix, the PVP-CDP ratio is increasing. In addition, it is also known from literature that the viscosity of PVP increases significantly with increasing PVP concentration, and increasing the polymer concentration in electrospinning solutions increases the fiber diameter. These discussions along with literature references have been added to the main body.

8. line 103; please explain what is the role of PVA and PVP in the electrospinning process.

A: It has been amended that the role of the PVA and PVP are to increase viscosity of the solution, making the electrospinning process possible.

9. line 103-104; how to ensure there is no CDP at other concentration in the nanofiber? please clarify if there is CDP embedded into the fibers?

A: The statement has been corrected to “the nanoparticle formation on the electrospun fiber surface was observed” to reflect the discovery. Several SEM images of a cross-sections of CDP decorated fibers have been added to the manuscript and to the Supplementary Material. These images show that indeed CDP can be embedded into the fibers. This has been added to the

manuscript body, and also analysed in the discussions.

10. line 127-130; the written text "The electrospun composite..." is more appropriate to be removed to the Methods section.

A: These lines have been moved to the Methods section.

11. line 137-138. the author mentioned the cell voltage is increased due to the triple phase boundary. this is not justifiable based on the test performed in this study.

A: This has been corrected in the manuscript in order to be clear on the connection with the triple phase boundary. The current statement stands as the following: "The increased fuel cell performance is likely due to the increase in available surface area for the oxygen reduction reaction at the triple phase boundary."

12. The discussion section needs to be revised significantly before it can be further for publication. The depth of discussion and comparison with state-of-the-art literature data is absent, making it difficult to determine whether significant advances have been made, or new insights gained.

A: This is a good point that lead us to increase the depth of the discussion, and more state-of-the-art literature data has been added to highlight the novelty of the reported findings.

13. line 208; the tube furnace mentioned does not look alike with the Fig. S4, which is a box furnace. please check.

A: Thank you for noticing the error. It has been corrected in the text. Indeed, a Barnstead box furnace was used for making the samples for the fuel cell studies.

14. line 212; typo of "Fig. SX"

A: The typo has been corrected.

15. line 213; there is not Fig. S8 in the appendix.

A: Thank you, the Figure has been added to the Supplementary material.

15. Please add in the method of Pt deposition in the Methods section.

A: The main information and the citation to the detailed Pt deposition method has been added to the Methods section.

Reviewer #3 (Remarks to the Author):

Review for manuscript: NCOMMS-18-03038

Title: Spontaneous formation of nanoparticles on electrospun fibers for fuel cell electrodes

This manuscript describes the fabrication of electrospun nanofibers by nozzle-free electrospinning and the spontaneous formation of cesium dihydrogen phosphate (CDP) nanoparticles onto their surfaces under specific experimental conditions. The generation of the CDP nanoparticles resulted to an enhanced electrocatalytic performance of the heat-treated membrane decorated with Pt catalytic nanoparticles, owned to the increased surface area of the electrospun CDP/polymer nanocomposite fibers.

The presented work is interesting. However, there are some points that should be addressed by the authors before considering publication in Nature Communications.

Major comments

- During the electrospinning process, the produced fibers were deposited on a substrate that was heated at 120 oC. Does the temperature of the heated substrate plays a key role in the spontaneous formation mechanism of the CDP nanoparticles? Would it be possible to control/tune the CDP nanoparticle diameters by varying this temperature?

A: The heating is needed for making it possible to electrospin the solution at low (40 mg/mL PVP and below) concentrations, but it doesn't seem the affect the spontaneous formation mechanism. We have obtained similar results with higher PVP concentration (80 mg/mL) at ambient temperature, using both a needleless and a conventional needled electrospinning apparatus. This has been added to the manuscript, see main body and the new section in supplementary information about these results.

- Why in the case of PVP, upon increasing the PVP concentration the nanoparticle density increases (lines 89, 90), while in the case of PVA the opposite is observed? (see lines 108, 109 – where it is stated that the nanoparticle density was higher for the lower PVA concentration).

A: Thank you for your observation. Indeed, the nanoparticle density shows opposite trends. A possible explanation has been added to the manuscript above Figure 3.

- Additional experimental data (i.e. SEM images) should be provided for the CDP-decorated fibers (a) after heat treatment at 300 °C for 12 hrs and (b) after Pt deposition. Does the heating step affect the CDP sizes and size distribution? What is the average size and size distribution of the deposited Pt nanoparticles?

A: Additional SEM images have been added to show the sample structure before and after heat treatment for both PVA and PVP, and after MOCVD. The MOCVD procedure strictly followed a procedure reported in a published paper, where it was stated that for the used Pt loading the Pt particles have a diameter of 3.7 nm, and are uniform. The size of the Pt nanoparticles have been added to the main manuscript, to the Methods section.

- In Figure 5, do the 2 graphs derive from single measurements? If not, error bars must be included. Is there a statistically significant difference between the 2 plots? Is there any information related to the electrocatalytic performance of the PVA-CDP analogues?

A: The 2 graphs in Figure 5 did derive from single measurements, as these measurements take a long time. The electrocatalytic performance from the PVA-CDP analogues are presented in Supplementary Fig. 11.

- In the discussion part, the authors should give more experimental evidence to support their statements related to the spontaneous formation mechanism of the CDP nanoparticles on the fibers' surfaces. Why this spontaneous process can only occur at a specific salt concentration? Is there any evidence of the existence of nanopores onto the fibers' surfaces that – according to the authors – may be involved in the CDP formation mechanism?

A: Additional experimental evidence has been added that shows the cross-section of a CDP-decorated electrospun fiber and elemental analysis has been also performed on the sample. Existence of nanopores has been observed, and an SEM image has been added as Supplementary Fig. 14. We believe that the nanoparticles appear on the surface by diffusion. Extra studies have been performed on the nanoparticle evolution over time, and humidity effects of the nanoparticle formation. The results have been added to the manuscript/supplementary information. It is not understood why the spontaneous nanoparticle formation can occur only at a specific salt concentration.

- In Figure S2, do the peaks observed for the CDP nanoparticle crystals correspond to monoclinic CsH_2PO_4 ? In the XRD spectrum provided for the

CDP nanocrystals I would also suggest to include the XRD spectrum of Pt-deposited analogue.

A: Thank you for your comment. Yes, it corresponds to the monoclinic form, as CDP is monoclinic at room temperature. It becomes cubic above 220 °C, and it returns to monoclinic structure when cooled back to room temperature. The analysis of the peaks before and after the MOCVD has been added to the supplementary information.

Minor comments

- In the introduction part, (page 3, lines 58-62), the authors mention several ways employed for the anchoring of nanoparticles onto the surfaces of electrospun fibers, including their immersion in solutions of dispersed nanoparticles, the in situ reduction of an appropriate precursor at their surfaces or the use of hydrothermal processes. Electrospaying is another way of depositing nanoparticles onto the surfaces of electrospun fibers and it can be performed simultaneously with electrospinning therefore avoiding time-consuming and multi-step procedures...A few references related to the use of electrospaying in the deposition of nanoparticles onto the surfaces of electrospun fibers should be included.

A: Thank you for the comment. Simultaneous electrospay with electrospinning indeed can result in nanoparticle decorated nanofibers as well, but that technique has several limitations. It has low yield, can experience clogging, as electrospay needs a nozzle, and the nanoparticles can't be as dense and homogeneous, as in the presented electrospinning method.

- In Figure 2, the presence of CDP nanoparticles is not obvious in the case of the 60 mg/mL PVP concentration.

A: Thank you for the comment. The image quality has been improved, so the nanoparticles of CDP are visible.

- Line 172. Please provide the solubility of the polymers in water.

A: The solubilities of the polymers in water have been added

- Line 212. Figure SX. Please provide the correct Figure numbering.

A: Thank you for the comment. Figure number has been added.

- Line 212. Figure S8 – there is no Figure S8.

A: Thank you for the comment. Figure S8 has been added.

- “Electrospray deposition” should be removed from keywords.

A: “Electrospray deposition” has been removed from keywords, and “electrospinning” has been added.

Reviewers' comments:

Reviewer #1 (Remarks to the Author):

Comments

The authors tried hard to explain and revise their manuscript. As this reviewer commented in the first round reviewing, the present work donot warrant its novelty at all for Nature Commun.

This reviewer would suggest to invite another reviewer who is qualified to judge the novelty and quality of this manuscript.

Their arguement "The reported method for fabricating nanoparticle decorated nanofibers in single-step electrospinning process shows promising potential for energy, catalysis or sensing applications" is not true if you search the references.

Reviewer #3 (Remarks to the Author):

The authors significantly revised their manuscript based on my initial comments. Therefore I would suggest publication of this work in Nature Communications.

“Reviewer #1 (Remarks to the Author):

Comments

The authors tried hard to explain and revise their manuscript. As this reviewer commented in the first round reviewing, the present work donot warrant its novelty at all for Nature Commun.

This reviewer would suggest to invite another reviewer who is qualified to judge the novelty and quality of this manuscript.

Their arguement "The reported method for fabricating nanoparticle decorated nanofibers in single-step electrospinning process shows promising potential for energy, catalysis or sensing applications" is not true if you search the references.”

A: Thank you very much for the additional time to review our paper one more time. Your critical comments made us work hard to explain and revise the entire manuscript. We acknowledge that the novelty of the findings was not highlighted enough. Since we received your comments after the second rounds of reviews, we have substantially revised the paper again, and we made even clearer the novelty of our findings, and even changed the title of the manuscript. The quality of the manuscript has been also significantly improved. We still believe that our results our novel. There is no published work that discusses our idea.

We offer two significant points to support our novelty claim:

1. The process of spontaneous formation of nanoparticles on electrospun nanofibers is new. Extensive published literature and patent searches produce no hits to prior disclosures. If you believe you know of one, please name it.
2. The application of electrospun fibers in solid acid fuel cell electrodes is also novel—we disclose it for the first time in a patent application. Please point out to even a single other prior reference.

We appreciate the recommendation to the editor to invite another opinion on the novelty of our work.

We also would like to point out that the offending sentence "The reported method for fabricating nanoparticle decorated nanofibers in single-step electrospinning process shows promising potential for energy, catalysis or sensing applications" is not stated in the manuscript. This sentence was written only in our answer to you to provide a general outlook for this technique. We believe that there is a need for a simple and inexpensive method for fabricating nanoparticle decorated nanofibers. Ref.14 in the revised manuscript describes how electrospun ceramic nanofibers with noble-metal particles could be used for catalytic applications, while Ref.15 reports on the use of a graphene decorated electrospun fiber structure for biosensing. Ref. 19 reports on the application of gold nanoparticle decorated electrospun fibers for photocatalysis, while Ref. 29 is our own patent application. All of these references support promising applications but **none** discloses the fabrication of nanoparticle decorated nanofibers in a single-step electrospinning process.

“Reviewer #3 (Remarks to the Author):

The authors significantly revised their manuscript based on my initial comments. Therefore I would suggest publication of this work in Nature Communications.”

A: Thank you very much for taking the time to check our revised manuscript and for recommending it for publication.